# Exploring the Biocontrol Potential of *Phanerochaete chrysosporium* against Wheat Crown Rot

**DOI:** 10.3390/jof10090641

**Published:** 2024-09-07

**Authors:** Lei Liu, Yaqiong Jin, Huijuan Lian, Qianxi Yin, Hailei Wang

**Affiliations:** College of Life Sciences, Henan Normal University, Xinxiang 453007, China; l-lei@whu.edu.cn (L.L.); 15515764519@163.com (Y.J.); 18303837104@163.com (H.L.); 18703733705@163.com (Q.Y.)

**Keywords:** wheat crown rot, *Fusarium pseudograminearum*, *Phanerochaete chrysosporium*, biological control, transcriptomes

## Abstract

The worldwide occurrence of wheat crown rot, predominantly caused by the pathogen *Fusarium pseudograminearum*, has a serious impact on wheat production. Numerous microorganisms have been employed as biocontrol agents, exhibiting effectiveness in addressing a wide array of plant pathogens through various pathways. The mycelium of the white-rot fungus *Phanerochaete chrysosporium* effectively inhibits the growth of *F. pseudograminearum* by tightly attaching to it and forming specialized penetrating structures. This process leads to the release of intracellular inclusions and the eventual disintegration of pathogen cells. Furthermore, volatile organic compounds and fermentation products produced by *P. chrysosporium* exhibit antifungal properties. A comprehensive understanding of the mechanisms and modalities of action will facilitate the advancement and implementation of this biocontrol fungus. In order to gain a deeper understanding of the mycoparasitic behavior of *P. chrysosporium*, transcriptome analyses were conducted to examine the interactions between *P. chrysosporium* and *F. pseudograminearum* at 36, 48, and 84 h. During mycoparasitism, the up-regulation of differentially expressed genes (DEGs) encoding fungal cell-wall-degrading enzymes (CWDEs), iron ion binding, and mycotoxins were mainly observed. Moreover, pot experiments revealed that *P. chrysosporium* not only promoted the growth and quality of wheat but also hindered the colonization of *F. pseudograminearum* in wheat seedlings. This led to a delay in the development of stem base rot, a reduction in disease severity and incidence, and the activation of the plant’s self-defense mechanisms. Our study provides important insights into the biocontrol mechanisms employed by *P. chrysosporium* against wheat crown rot caused by *F. pseudograminearum*.

## 1. Introduction

Wheat (*Triticum aestivum* L.) is a prominent grain crop on a global scale, holding the second highest position in terms of importance. China, being the largest producer and consumer of wheat, accounted for approximately 20% of its total grain output in 2023. As a result, wheat plays a crucial role in safeguarding food security in China. The management of wheat diseases is crucial for maintaining high yields and quality of wheat [1]. Wheat crown rot (WCR) is a significant and widespread soil-borne disease affecting wheat in numerous key wheat-producing areas globally, with substantial implications for wheat quality and yield [2]. The regular practice of reincorporating wheat and corn residues into the soil and the implementation of continuous cropping techniques contribute to the survival of pathogens throughout the winter and summer seasons, leading to a notable increase in pathogen populations within the soil [3]. At present, there is still a deficiency in the availability of wheat cultivars that demonstrate strong resistance to wheat crown rot during cultivation.

The prevalence of wheat crown rot is primarily associated with species within the *Fusarium* genera, specifically *F. pseudograminearum*, *F. graminearum, F. culmorum, F. avenaceum*, and *F. oxysporum* [4,5,6]. This pathogen exhibits a strong survival ability in soil stubble tissue during the winter months, existing in the forms of conidia, mycelium, and chlamydospores, with a potential longevity of up to three years. In conducive environments, a substantial quantity of asexual macroconidia can be generated, leading to plant infection. The disease can present in wheat at various growth stages, from seedling to maturity, as noted by Kazan and Gardiner [7]. Excessive field moisture may lead to the formation of pink or white mold layers on the stem nodes and internodes, ultimately resulting in the production of shriveled or absent grains. Furthermore, the susceptibility of wheat stems to breakage contributes to premature lodging and plant death. It is imperative to acknowledge the clear correlation between disease outbreaks and agricultural output. Moreover, the secondary metabolites generated by *F. pseudograminearum* in the course of plant invasion, particularly the type-B trichothecene toxins (3-acetyldeoxynivalenol and 15-acetyldeoxynivalenol), have a substantial impact on food safety and the onset of disease outbreaks [8]. A case in point is China’s documentation in 2011 of wheat stem rot caused by *F. pseudograminearum* [9], particularly in the northern Henan province region, where the resulting yield losses ranged from 30% to 50%.

The utilization of biological control methods offers a sustainable approach to managing harmful organisms, reducing dependency on synthetic pesticides, and mitigating environmental pollution [10,11]. Additionally, it shows potential for improving crop quality and increasing economic returns for farmers, making it a viable strategy for preventing and controlling stem rot diseases. The biological control of plant diseases typically involves the application of microorganisms or their byproducts to suppress disease development [12]. Several microorganisms are currently under investigation for their potential in the prevention and management of soil fungal diseases. The examination of the interaction between antagonistic microorganisms and indigenous soil pathogens is essential in the development of more efficient biological control methods. These mechanisms encompass various strategies, including fungal mycoparasitism, antibiotic synthesis, competition for nutrients, the induction of plant resistance, and the promotion of plant growth [13]. However, due to the instability in field applications and the inconvenience of storage and transportation, only species in some genera, such as *Bacillus*, *Pseudomonas*, *Streptomyces*, *Trichoderma*, *Coniothyrium*, and *Gliocladium*, have been commercialized [14]. Prior studies have documented the inhibitory impacts on *F. pseudograminearum* of *Phoma mediterranea* [15], *Streptomyces* isolates [16], *T. afroharzianum*, *T. harzianum*, *T. gamsii* [17], and *Rhizophagus intraradices* [18].

*Phanerochaete chrysosporium* is a typical representative of white-rot fungi belonging to Phanerochaetaceae under Polyporales, Agaricomycetidae, and Basidiomycota. Its substantial extracellular peroxidase production and lack of specific substrate requirements render it suitable for the degradation of a diverse array of organic compounds [19], in addition to its lignin-degradation capabilities [20]. Previous studies have shown that the rapid decomposition of crop residue containing pathogens, facilitated by the dense planting of *Brassica canopies*, may play a crucial role in alleviating wheat crown rot [21]. Pradeep et al. [22] illustrated the efficient breakdown of crop residues by *P. chrysosporium*, indicating a potential decrease in pathogen survival. Other researchers have demonstrated that *P. chrysosporium* is capable of efficiently degrading phenolic acids in soil under conditions of continuous cropping, leading to a reduction in the occurrence of cucumber root diseases and pests [23]. Additionally, the efficacy of *P. chrysosporium* in mitigating cut chrysanthemum wilt disease caused by *F. oxysporum* has been confirmed [24]. At present, there is a notable deficiency in research dedicated to improving the management of wheat crown rot via the application of white-rot fungi to suppress *F. pseudograminearum*. This study aims to employ *P. chrysosporium* as a biocontrol agent and assess its efficacy in controlling wheat crown rot and its impact on wheat growth. The findings from this investigation are expected to offer novel strategies and biocontrol approaches for mitigating fungal diseases in plants.

## 2. Materials and Methods

### 2.1. Microorganism

*P. chrysosporium* (ATCC20296) was acquired from the Research Center for Eco-Environmental Sciences, Chinese Academy of Sciences, China, and *F. pseudograminearum* was received as a donation from the Henan Institute of Science and Technology. The fungi demonstrated growth on potato dextrose agar (PDA) at a temperature of 28 °C and were preserved at a temperature of −80 °C in a solution containing 30% glycerol.

### 2.2. Dual Culture Test and Observation of Mycelium Interaction

A plate confrontation assay was utilized to assess the antifungal activity of *P. chrysosporium* against *F. pseudograminearum* associated with wheat crown rot in China. *F. pseudograminearum* and *P. chrysosporium* were inoculated on both sides of a PDA Petri dish (φ 90 mm) in the experimental group, while plates with only *F. pseudograminearum* served as the control group according to the guidelines established by Zhao et al. [25], with each treatment replicated thrice and subjected to incubation at 28 °C. After a 6-day co-cultivation, the level of colony interaction between *F. pseudograminearum* and *P. chrysosporium* was evaluated. The inhibition rate was determined by applying the formula (C−T)×100%/C, where C denoted the colony diameter of the control and T denoted the colony diameter of the treated sample, as described by Dhanabalan et al. [26].

The point of intersection between two different hyphae was carefully excised with a blade, allowing for the observation and documentation of hyphal growth and interaction using an inverted light microscope (Leica DMi1, Leica Microsystems, Wetzlar, Germany). Simultaneously, aseptic cellophane was centrally positioned on a PDA plate, with fungal cultures of *P. chrysosporium* and *F. pseudograminearum* situated at opposite diagonal extremes of the cellophane. Hyphal samples from the confrontation region were gathered at designated time intervals (48 h, 60 h, 72 h, 84 h, and 120 h) for the examination of hyphal interactions utilizing the Leica TCS SP8 laser scanning confocal microscope (LSCM) [27].

### 2.3. Fungal Volatile Organic Compound (VOC) Antifungal Experiment

*P. chrysosporium* was inoculated on a PDA Petri dish (φ 90 mm) for 2 days in advance to be able to produce enough VOCs and then placed on a divided dish with *F. pseudograminearum* separated by cellophane. Then, inoculated dishes were sealed with parafilm and incubated at 28 °C for 5 days according to Song et al. [28]. The growth of both organisms was monitored under a scanning electron microscope (SEM) after 3, 4, and 5 days, and the antifungal efficacy was determined. Two vertical diameters of each colony were measured at the point where the fungal mycelia of the control colony reached the edge of the Petri dish. Additionally, *F. pseudograminearum* sealed with a PDA plate was observed, with another *F. pseudograminearum* colony sealed with the same species serving as the control group. The experiment was repeated three times, and the inhibition rate was determined, consistent with the previous description.

### 2.4. Fungal Inhibition Experiment of Fermentation Products of P. chrysosporium

*P. chrysosporium* fermentation broth was prepared according to Chai et al. [29] and was added to PDA solid medium at final concentrations of 0 g/mL, 0.1 g/mL, 0.3 g/mL, and 0.5 g/mL. *F. pseudograminearum* colonies with 6 mm diameters were then inoculated onto the center of these plates. To further investigate the inducement effect of *F. pseudograminearum* cell wall components on the inhibitory activity of *P. chrysosporium*, a 4 g/L preparation of *F. pseudograminearum* cell walls was introduced into fermentation broth of *P. chrysosporium* that had been cultured for 24 h and incubated at 28 °C for plate inhibitory tests according to Yang et al. [30]. PDA plates without *P. chrysosporium* fermentation broth were used as the control. Each treatment was replicated three times. The colony diameter of *F. pseudograminearum* was measured on days 3, 4, and 5 and compared to the control to calculate the inhibition rate. The activity of cell-wall-degrading enzymes (CWDEs) in the induction fermentation broth involved in the degradation of the cell wall of pathogen was evaluated through the utilization of the 3,5-dinitrosalicylic acid method [31], with a focus on quantifying the activities of proteinase, endoglucanase, chitinase, and beta-1,3-glucanase.

### 2.5. Transcriptome Analysis of Interaction between P. chrysosporium and F. pseudograminearum in Different Confrontation Periods

*P. chrysosporium* and *F. pseudograminearum* cakes were prepared and inoculated at both ends of the PDA medium covered with cellophane with a 6 cm interval. Furthermore, one side of the medium was inoculated solely with *P. chrysosporium* to serve as the control group. All cultures were maintained in a constant-temperature incubator set at 28 °C. During the confrontation experiment, fungal mycelia were gathered from the surrounding area of the contact zone at three specific time points: prior to contact (36 h), upon initial contact with the mycelium (48 h), and during the subsequent overgrowth of *P. chrysosporium* on *F. pseudograminearum* mycelia (84 h). Concurrently, mycelia from the advancing front of growth of *P. chrysosporium* and *F. pseudograminearum* were harvested at each designated time point. Three replicates were established for each time point. The harvested mycelia were transferred to 1.5 mL centrifuge tubes, promptly frozen using liquid nitrogen, and preserved at −80 °C for subsequent experiments.

Samples were sent to Shanghai Mejisen Biomedical Technology Co., Ltd. (Shanghai, China) for sequencing using the Illumina Novaseq 6000 platform (San Diego, CA, USA). Raw data were quality controlled to obtain clean data. Using HISAT2 software version 2.2, quality control data were aligned with the *P. chrysosporium* reference genome GCA000167175.1 (http://fungi.ensembl.org/Phanerochaete_chrysosporium/Info/Index, accessed on 1 December 2022), and the distribution of MappedReads was analyzed. The genes/transcripts obtained were aligned with six prominent protein databases (GO, KEGG, COG, NR, Swiss-Prot, Pfam), and the derived functional annotation information was subjected to analysis. The gene expression levels of clean reads were quantitatively assessed using the expression quantification software RSEM version 1.3.3, with TPM (transcripts per million reads) serving as the expression measurement index. Venn analysis was performed on gene expression levels in various samples to identify commonly and specifically expressed genes. DESeq2 version 3.11 was utilized to analyze ReadCounts data from the gene expression analysis, identifying differentially expressed genes (DEGs) with Padjust < 0.05 and |log_2_ FC| ≥ 1 between groups at different time points [32]. Differential gene expression at 36 h, 48 h, and 84 h of confrontation, as well as their corresponding time-point single-cultured samples, were analyzed. The differential genes identified were then subjected to Gene Ontology (GO) enrichment analysis using the Gotools version 4.5 software, utilizing Fisher’s exact test and setting a threshold of Padjust < 0.05 to control for false positives in the analysis.

RNA was extracted utilizing the EzgeneTM Fungal RNA Miniprep kit (Biomiga, San Diego, CA, USA), followed by cDNA synthesis of 50 ng RNA using the Omniscript RT kit (Qiagen, Hilden, Germany), as per the manufacturer’s guidelines. A total of 24 genes were selected for the validation of RNA-seq differential gene expression data through qRT-PCR, with primers designed using Primer version 5.0 based on the assembled transcriptome for amplification. The primers employed were documented in Appendix A. The glyceraldehyde-3-phosphate dehydrogenase gene (gpd) served as an internal reference gene, and the relative gene expression levels were determined utilizing the comparative Ct method with the equation 2^−ΔΔCt^ [33]. RT-qPCR analyses were conducted in triplicate with three biological replicates. Subsequently, the RT-qPCR outcomes were juxtaposed with the transcriptome data to ascertain the correlation of each gene’s expression.

### 2.6. Pot Control Effect of P. chrysosporium on Wheat Stem Rot

#### 2.6.1. Influence of *P. chrysosporium* on Growth of Wheat 

We chose plump, uniform seeds free from visible signs of mold and subjected them to a 2 min immersion in a 0.6% NaClO solution, followed by two rinses with sterile water. Subsequently, the seeds were immersed in 75% ethanol for 2 min, rinsed three times with sterile water, and allowed to soak in sterile water for 12 h. The soaked seeds were planted in a culture medium sterilized via high-temperature treatment, consisting of a peat and vermiculite mixture in a ratio of 3:1. Each pot was filled with soil to a level 3 cm below the rim, and 11 seeds were sown per pot, with 5 pots allocated for each treatment (26 °C, 60% humidity, 12 h light/12 h dark cycle, 8000 lux). The experimental group was sprayed with a *P. chrysosporium* spore solution (1 × 10^7^ spores/mL dispersed in 0.1% Tween-20) on both the soil and the entire wheat plants according to Hasan et al. [34], with the use of a 0.1% Tween-20 solution as a blank control. The plant height and fresh weight of the aboveground part of wheat plants were measured at 15 days post-inoculation (dpi). Wheat leaf tissues treated with different methods were collected at 1, 3, 5, and 7 dpi. Samples from each treatment group were assessed for root viability using the 2,3,5-triphenyl tetrazolium chloride (TTC) method as described by Wang et al. [35], and the chlorophyll content was determined following the protocol outlined by Bahrami et al. [36]. The strain’s ability to secrete indole-3-acetic acid (IAA) was determined using the Salkowski colorimetric method [37].

#### 2.6.2. *P. chrysosporium* Enhancement of Wheat Growth under *F. pseudograminearum* Stress

The nutrient soil mixture, comprising equal parts of soil, sandy soil, and peat soil (1:1:1 ratio), was meticulously mixed and sterilized at 121 °C for 2 h. Wheat seeds were rigorously selected according to specific criteria. Upon reaching the one-leaf–one-heart stage—defined by the complete expansion of the first leaf and the emergence of the second leaf with a rolled leaf blade—the wheat plants were subjected to treatment [38]. For analytical purposes, each treatment included 5 pots, with 11 seeds sown per pot. A control group was treated with water containing 0.2% Tween-20, while four different treatments included (1) inoculation with *F. pseudograminearum* only, (2) inoculation with *F. pseudograminearum* followed by carbendazim spray for the control, (3) inoculation with *F. pseudograminearum* and treatment with *P. chrysosporium* spore suspension spray (1 × 10^7^ spores/mL), and (4) only treatment with *P. chrysosporium* spore suspension spray (1 × 10^7^ spores/mL). Each treatment had three replicates. *F. pseudograminearum* was introduced through the inoculation of diseased wheat grains, with two grains placed at the base of each wheat seedling according to Zhang et al. [39].

The disease status of wheat was assessed at 25 dpi. Wheat seedlings were carefully uprooted, the soil was rinsed off, and the severity of stem base rot was graded on a scale of 0–5 by referring to Alkher et al. [40] based on the condition of the plant leaves as follows: level 0 indicating no visible symptoms, level 1 representing leaf sheath browning less than 10% of the length, level 2 indicating browning of the first leaf sheath accounting for 11% to 25% of its length, level 3 representing browning of the first leaf sheath accounting for 26% to 50% of its length, level 4 indicating obvious browning of the second leaf sheath, and level 5 representing either obvious browning of the third leaf sheath or plant death. The disease index (DI) = Σ (number of plants in each disease level × corresponding level)/(total number of plants surveyed × highest level) × 100% was used to represent the severity of illness. The relative control efficacy = (control disease severity − treatment group disease severity)/control group disease severity × 100%. The activities of superoxide dismutase (SOD), catalase (CAT), and peroxidase (POD) in seedlings were quantified using kits from Grace Biotechnology, Suzhou, China, following the manufacturer’s instructions [41].

The total RNA of wheat seedlings was extracted utilizing a plant total RNA extraction kit obtained from Magen. The BioGlodTM Script Plus All-in-one 1st Strand cDNA Synthesis SuperMix (+gDNA wiper) kit was employed for the synthesis of first-strand cDNA using total RNA as a template, following the manufacturer’s instructions. Primers targeting five genes, including antioxidant enzyme-related genes and defense-related genes, were designed using Primer premier 5.0 primer design software and synthesized by Shanghai Biotechnology Co., Ltd., Shanghai, China (Appendix A). The synthesized cDNA served as the template for fluorescence detection utilizing the Talent fluorescence quantitative detection kit (SYBR Green) FP209 reagent in conjunction with the Roche Light Cycler^®^ 96 fluorescence quantitative PCR instrument (Roche Diagnostics GmbH, Mannheim, Germany). The relative gene expression was determined through the application of the 2^−ΔΔCt^ formula.

#### 2.6.3. Quantification of *F. pseudograminearum* Colonization under *P. chrysosporium* Treatments

In potting prevention experiments, *F. pseudograminearum* expressing green fluorescence (pCT74-HygB-GFP) was generated and employed as the inoculum, as described in [42]. The fungus was cultured on PDA medium supplemented with hygromycin to select for the resistance gene, and its identity was verified through morphological assessments. Wheat seedling tissues obtained at 10, 15, and 25 dpi were processed following established protocols. Fungal isolates were obtained, and colonies were visually inspected and enumerated thereafter.

At 25 dpi, wheat seedling tissues from each group (no treatment, *F. pseudograminearum* treatment, *F. pseudograminearum* and *P. chrysosporium* treatment) were collected and subjected to the removal of soil from the roots and stem bases through rinsing with running water. Subsequently, 1 cm stem segments were randomly excised from the stem bases and roots near the stem bases, and the presence and distribution of fluorescently labeled *F. pseudograminearum* within the tissues were visualized using LCSM.

The DNA fragment of *F. pseudograminearum* from pure culture was amplified using the specific primers FP-4F (5′-GTGTCAATCAGTCACTAACAACC-3′) and FP-4R (5′-GAGGACAATAGTGACAACATACC-3′), resulting in a target fragment of 194 bp. Following amplification, the target fragment DNA was quantified using a microspectrophotometer. The quantified DNA served as a standard for gradient dilution, resulting in the establishment of a standard curve with 6 gradients ranging from 10^−1^ to 10^−6^ ng/µL. A 2 cm tissue sample was excised from the base of the wheat stem, placed in a pre-cooled mortar with an appropriate amount of liquid nitrogen, ground into powder, and transferred to a 1.5 mL centrifuge tube. Total DNA was extracted from the wheat seedling stem base tissue using the HiPure Fungal DNA Kit (Magen, Guangzhou, China) (100 mg). The extracted genomic DNA served as a template for fluorescence detection using the Talent FP209 assay (SYBR Green) kit (Mei5bio, Beijing, China) in conjunction with the Roche Light Cycler^®^ 96 fluorescence quantitative PCR instrument following the instruction. The Light Cycler ^®^ 96 software version 1.1 automatically generated the standard curve and absolute quantitative results.

### 2.7. Statistical Analysis

The data were standardized using Excel 2019, followed by a one-way analysis of variance (ANOVA) conducted using SPSS 22.0 (SPSS Inc., Chicago, IL, USA). Significance analysis was carried out using a *t*-test, with statistical significance indicated by *p* < 0.05, *p* < 0.01, and *p* < 0.001.

## 3. Results

### 3.1. Tablet Confrontation Inhibition between P. chrysosporium and F. pseudograminearum

With an extended culture time, rapid growth of both the *F. pseudograminearum* mycelia and *P. chrysosporium* was observed on PDA plates, as depicted in (Figure 1A,B), respectively. It is evident that the latter exhibited a higher growth rate compared to the former. On the co-cultured plate, *P. chrysosporium* mycelia began to contact *F. pseudograminearum* at 48 h, subsequently leading to a significant inhibition of *F. pseudograminearum* mycelial growth. After contact, the growth of *F. pseudograminearum* was observed to decelerate, accompanied by pigment deposition in the confrontation zone (Figure 1C,D). By the fifth day of co-cultivation, a significant inhibitory effect was observed, with a 47% inhibition rate (Figure 1E). These findings suggested that, in the direct confrontation experiment, *P. chrysosporium* exerted obvious inhibitory actions (*p* < 0.001) on *F. pseudograminearum* through its rapid mycelial growth, effectively occupying survival space.

The mycelium diameters of *P. chrysosporium* and *F. pseudograminearum* were measured in culture. *F. pseudograminearum* had a mean diameter of 3.706 µm, while *P. chrysosporium* had a mean diameter of 1.328 µm. There was a significant difference in mycelial diameters between the two species (*p* < 0.001) (Figure 2A–C). Following 48 h of co-culturing, the mycelium of *P. chrysosporium* initiated contact with the mycelium of *F. pseudograminearum*, then entwining and adhering to the latter (Figure 2D). Subsequently, from 60 h to 84 h of co-culturing, a notable increase in intertwined areas between the mycelia of *P. chrysosporium* and *F. pseudograminearum* was observed in the confrontation zone, accompanied by the growth and entwining of the *P. chrysosporium* mycelia on the *F. pseudograminearum* mycelium, as well as the emergence of an adsorption structure (Figure 2E) and the prolific production of chlamydospores by *P. chrysosporium* to compete for growth space with *F. pseudograminearum* (Figure 2F). Following 120 h of co-culturing, a notable growth inhibition of *F. pseudograminearum* by *P. chrysosporium* was observed, and mycelial rupture gradually appeared in *F. pseudograminearum*, resulting in the release of its contents into the surrounding culture medium (Figure 2G–I).

### 3.2. Effect of VOCs Produced by P. chrysosporium on Pathogen Growth

After inoculated dishes were sealed with parafilm and incubated for 3 days, two vertical diameters of each colony were measured. The colony growth diameter of *F. pseudograminearum* with *P. chrysosporium* was 3.431 ± 0.039 cm, compared to 4.36 ± 0.042 cm and 4.467 ± 0.036 cm for the colony growth diameters with PDA plates and *F. pseudograminearum*, respectively. Similar observations were made on the fourth (4.35 ± 0.075 cm, 6.001 ± 0.033 cm, 6.047 ± 0.157 cm) and fifth days (5.414 ± 0.018 cm, 7.162 ± 0.091 cm, 6.861 ± 0.200 cm) of the experiment (Figure 3A). The difference between the *P. chrysosporium* treatment group and the latter two groups was found to be statistically significant (*p* < 0.001) (Figure 3B), while the difference between the latter two groups was not statistically significant. These findings suggested that the VOCs produced by *P. chrysosporium* had a suppressive impact on the growth of *F. pseudograminearum*.

SEM was employed to examine the morphology of both inhibited and control-group hyphae. The findings revealed that exposure to VOCs produced by *P. chrysosporium* resulted in a reduction in hyphal thickness (Figure 3C,D), suggesting a significant influence of *P. chrysosporium*-produced VOCs on the growth of *F. pseudograminearum*.

### 3.3. Inhibition of F. graminearum by P. chrysosporium Fermentation Products

After adding a *F. pseudograminearum* cell wall preparation to the fermentation process of *F. pseudograminearum*, an increase in the antifungal activity of *P. chrysosporium* fermentation filtrate was observed, as shown in Figure 4A,B. At concentrations of 0.1 g/mL, 0.3 g/mL, and 0.5 g/mL, the inhibition rates of the *P. chrysosporium* fermentation products were 20.83%, 17.36%, and 16.67% respectively, while the inhibition rates in the group with cell wall preparation were 23.22%, 23.76%, and 26.19% respectively (Figure 4C). These data indicated that adding *F. pseudograminearum* cell wall preparation to the fermentation process of *P. chrysosporium* can induce the production of substances in the *P. chrysosporium* fermentation liquid, thus enhancing the inhibitory activity against *F. pseudograminearum*.

According to Figure 5, in comparison to that in the fermentation broth lacking cell wall preparation, the chitinase activity in the induced fermentation broth was notably elevated on the first and third days post-addition, with the highest chitinase activity recorded at 1.89 U/mL on the first day (Figure 5A). Additionally, β-1,3-glucanase activity showed a significant induction on the third and fifth days, with the highest activity also observed on the first day at 59.44 U/mL (Figure 5B). Furthermore, the addition of *F. pseudograminearum* cell wall preparations resulted in a notable increase in endoglucanase activity, reaching a maximum of 0.64 U/mL on the first day (Figure 5C). The proteinase activity of *P. chrysosporium* exhibited a gradual increase over time following the addition of cell wall preparation of *F. pseudograminearum*, culminating in peak activity on the fifth day 0.15 U/mL (Figure 5D). These results indicated that the addition of *F. pseudograminearum* cell wall preparations to the liquid culture of *P. chrysosporium* induced the production of CWDEs (protease, chitinase, endoglucanase, and β-1,3-glucanase), thereby enhancing the activity of CWDEs in the fermentation broth.

### 3.4. Transcriptome Analysis of Interactions between P. chrysosporium and F. pseudograminearum at Various Confrontation Stages

From Figure 6A, it can be observed that there are fewer significantly differentially expressed genes encoding CWDEs (proteases, chitinases, and glucanases) of *P. chrysosporium* during co-cultivation at 36 h and 48 h. Specifically, at 36h, there were 17 significantly differentially expressed genes, with 4 significantly up-regulated genes and 13 significantly down-regulated genes. At 48 h, there were 10 significantly differentially expressed genes, all of which were significantly down-regulated. The number of significantly differentially expressed genes increased to 56 at 84 h of co-cultivation, comprising 37 significantly up-regulated genes and 19 significantly down-regulated genes. The expression levels of eight genes encoding proteases were significantly up-regulated (log_2_ FC (PcFp_84 h/Pc_84 h) > 2), with log_2_ FC (PcFp_84 h/Pc_84 h) > 3 for four genes, including *AGR57_10019*, *AGR57_6608*, *AGR57_4271*, and *AGR57_2089*. Furthermore, the expression levels of three genes encoding chitinases were significantly up-regulated (log_2_ FC (PcFp_84 h/Pc_84 h) > 1), with log_2_ FC (PcFp_84 h/Pc_84 h) > 2 for gene *AGR57_6505*. Additionally, the expression levels of 10 genes encoding glucanases were significantly up-regulated (log_2_ FC (PcFp_84 h/Pc_84 h) > 1), with log_2_ FC (PcFp_84 h/Pc_84 h) > 2 for gene *AGR57_7645*. Genes encoding CWDEs were significantly expressed in *P. chrysosporium* and *F. pseudograminearum* during the late stage of contact (84 h). Clustering analysis was conducted on the genes related to iron ion binding produced by *P. chrysosporium* during three different confrontation periods, and the results are shown in (Figure 6B). The number of genes significantly up-regulated in association with iron ion binding increased notably at 84 h of confrontation. At 84 h, there were a total of 51 DEGs related to iron ion binding, among which 11 were significantly down-regulated DEGs and 40 were significantly up-regulated DEGs. Among the up-regulated DEGs, five exhibited a log_2_ FC (PcFp_84 h/Pc_84 h) > 4 (*AGR57_15246*, *AGR57_14180*, *AGR57_7662*, and *AGR57_14178*). Through clustering analysis of genes related to the synthesis of mycotoxins in *P. chrysosporium* during three different confrontation periods (Figure 6C), a total of nine DEGs related to mycotoxin synthesis were identified after 84 h of co-cultivation. All nine DEGs were significantly up-regulated, with log_2_ FC (PcFp_84 h/Pc_84 h) > 2 for seven DEGs. Specifically, the log_2_ FC (PcFp_84 h/Pc_84 h) values for *AGR57_6105*, *AGR57_4396*, *AGR57_14274*, and *AGR57_14275* were 4.376, 3.810, 3.480, and 3.384, respectively.

After analyzing transcriptome data, the relative expression levels of nine DEGs were validated using quantitative real-time polymerase chain reaction (qRT-PCR). Among them, one gene was up-regulated in each confrontation period, *AGR57_4555* (hydrolase activity), and three genes were down-regulated in each confrontation period: *AGR57_10505* (polysaccharide catabolism process), *AGR57_7806* (hydrolase activity), and *AGR57_3705* (lipid metabolism process). Five genes were significantly up-regulated at 84 h: *AGR57_9511* (iron ion binding), *AGR57_8924* (containing jacalin-like lectin protein domain), *AGR57_8923* (containing jacalin-like lectin protein domain), *AGR57_3689* (hydrolase activity), and *AGR57_4271* (proteolysis). The qRT-PCR findings, depicted in the accompanying figure (Figure 6D), were in agreement with the DEG analysis results obtained from RNA-seq for the specified genes at various time points of confrontation.

### 3.5. P. chrysosporium’s Pot Control Effect on Wheat Stem Rot

#### 3.5.1. Impact of *P. chrysosporium* on Growth of Wheat

During the seedling stage, the plant height of wheat plants in the *P. chrysosporium* treatment group exhibited a 13.39% increase in comparison to the control group with 0.1% Tween-20, as depicted in Figure 7A. Additionally, the aboveground fresh weight of the treatment group showed a 17.76% increase relative to the control group, as illustrated in Figure 7B. Following treatment with *P. chrysosporium*, the root length exhibited a statistically significant increase of 17.63% compared to the control group (*p* > 0.001) (Figure 7C), while root viability relative to the control group increased by 23.54% (Figure 7D). Through the quantitative analysis of chlorophyll content, it was determined that there was no statistically significant variance in the levels of chlorophyll a and chlorophyll b between the treatment and control groups (Figure 7E,F). After subjecting the fermentation liquid of *P. chrysosporium* to reaction with Salkowski reagent in the absence of light, it was observed that the color of the liquid turned pink on the fifth day, suggesting the secretion of IAA by *P. chrysosporium*. Furthermore, the intensity of color in the reaction mixture of *P. chrysosporium* fermentation liquid exceeded that of the 50 mg/L IAA standard, indicating a minimum concentration of 50 mg/L of IAA in the fifth-day *P. chrysosporium* fermentation liquid (Figure 7G). Following this, a quantitative analysis of the IAA concentration in the fermentation liquid on the fifth, sixth, and seventh days indicated that the highest production of IAA occurred on the fifth day, reaching a concentration of 152.5446 mg/L (Figure 7H).

#### 3.5.2. *P. chrysosporium* Enhanced Resistance of Wheat Plants to Stem Base Rot

There was no sign of disease in the control group or the *P. chrysosporium*-only group, while the addition of *P. chrysosporium* considerably improved the growth status of wheat inoculated with *F. pseudograminearum* and indicated a reduction in the number of wheat plants exhibiting browning in the stem base and a better plant growth status (Figure 8A,B). Simultaneously, the severity, incidence rate, and relative control efficacy of wheat crown rot were assessed in accordance with established grading criteria (Figure 8C). Statistical analysis was conducted on each treatment group of *P. chrysosporium* at 25 dpi. The findings revealed that the incidence rates of the *F. pseudograminearum* group, the carbendazim treatment group, and the *P. chrysosporium* treatment group were 64.34%, 53.33%, and 34.16%, respectively. Following treatment with *P. chrysosporium*, the incidence rate of wheat crown rot exhibited a significant reduction of 20.18% compared to that in the *F. pseudograminearum* group. The severity of wheat stem base rot in the *F. pseudograminearum* group, the carbendazim treatment group, and the *P. chrysosporium* treatment group were 33.28%, 27.77%, and 14.9%, respectively. The efficacy of *P. chrysosporium* in controlling *F. pseudograminearum* was significantly greater than that of carbendazim, resulting in a relative increase in control efficacy of 38.66%. These findings indicated that the introduction of *P. chrysosporium* can effectively reduce the plant disease from *F. pseudograminearum* infection.

#### 3.5.3. *P. chrysosporium* Promoted Antioxidant Enzyme Activity and Enhanced Defense Genes Expression in Wheat Plants under Pathogen Stress

A notable influence on antioxidant enzyme activity in wheat seedlings by adding *P. chrysosporium* was observed under pathogen stress. The wheat seedlings treated with both *P. chrysosporium* and *F. pseudograminearum* exhibited significantly increases of 2.3%, 4.5%, and 45.41% in SOD, POD, and CAT activities, respectively, compared to those treated with only *F. pseudograminearum* (Figure 9A–C). Simultaneously, wheat seedlings exposed to a combined treatment with *P. chrysosporium* and *F. pseudograminearum* demonstrated a significant increase of 126.4% and 418.94% in the transcription levels of *SOD* and *POD* genes, respectively, relative to the control group. Furthermore, these levels were 87.88% and 106.3% higher than those observed in seedlings treated solely with *F. pseudograminearum*. (Figure 9D,E).

The transcription levels of pathogenesis-related protein 4 (*PR*4), pathogenesis-related protein 1.1 (*PR*1.1), chitinase (*CHI*), and β-1,3-glucanase precursor gene (*Glb*3) in wheat seedlings treated with *F. pseudograminearum* alone exhibited increases of 136.29%, 59.56%, 32.45%, and 90.5%, respectively, when compared to those in the control group. Moreover, higher transcription levels of *PR*4, *PR*1.1, *CHI*, and *Glb*3 were observed in wheat seedlings treated with a combination of *P. chrysosporium* and *F. pseudograminearum*, increased by 232.52%, 171.12%, 279.83%, and 298.62% when compared to those in the control group, respectively (Figure 9F–I).

#### 3.5.4. *P. chrysosporium* Reduced Pathogen Colonization Levels in Wheat

The quantification of the pathogenic fungus *F. pseudograminearum* in wheat tissues was conducted through the plating of root and stem segments for fungal isolation tests. The quantification of pathogenic fungi in the roots of the *F. pseudograminearum* group and the treatment group (*P. chrysosporium* + *F. pseudograminearum*) yielded counts of 650, 375, and 1000 and 150, 150, and 500 CFU/g of fresh weight (FW) at 10, 20, and 30 dpi, respectively. The counting of pathogens in the stem base in the *F. pseudograminearum* group and the treatment group yielded 25, 800, and 2000 and 0, 250, and 1000 CFU/g (FW), respectively (Figure 10A). The proliferation of pathogens within each group exhibited a positive correlation with time, consistent with the typical progression of diseases. Furthermore, the population of pathogens present in the root and stem base of wheat subjected to treatment with *P. chrysosporium* was notably diminished compared to that in the *F. pseudograminearum* group. These findings suggested that the introduction of *P. chrysosporium* led to a decrease in the colonization of *F. pseudograminearum* in the roots and stem bases of wheat.

In the potting experiment, root sections proximal to the stem base were extracted at 20 dpi and examined using LCSM to analyze the spatial distribution of *F. pseudograminearum* labeled with GFP + Hyg within the roots. The results depicted in Figure 10B indicated that in the control group, plant root cells exhibited a compact arrangement, with no discernible presence of green-labeled hyphae or conidial structures within the root tissues. In the experimental group that was inoculated with *F. pseudograminearum*, a significant presence of hyphae was observed within the intercellular spaces at the central region of the root, displaying a reticular distribution. Conversely, in the group treated with *P. chrysosporium*, the distribution of *F. pseudograminearum* hyphae was confined to the root surface cells, showing minimal inward extension, and an absence of *F. pseudograminearum* distribution within the intercellular spaces was found at the root center. These results indicated that the introduction and establishment of *P. chrysosporium* may competitively occupy ecological niches, thereby impeding the colonization of *F. pseudograminearum* in the outer layers of root cells and subsequently hindering the rapid invasion of plants via vascular bundles.

Changes in the DNA content of *F. pseudograminearum* in the root and stem base tissues of the *F. pseudograminearum* group and *P. chrysosporium* treatment group were detected using absolute quantitative PCR, reflecting the colonization levels of *F. pseudograminearum*. A high correlation coefficient (R^2^ > 0.99) was achieved in the establishment of a standard curve correlating DNA content and Cq values, meeting the necessary criteria for absolute quantitative PCR analysis. The standard curve equation, y = −5.0757x − 2.65 (R^2^ = 0.9904) was successfully determined. According to (Figure 10C), the DNA content of *F. pseudograminearum* in wheat roots at 10, 20, and 35 dpi was measured to be 1.88579 × 10^−6^ ng, 3.33027 × 10^−6^ ng, and 2.9667 × 10^−6^ ng, respectively. Similarly, the DNA content of *F. pseudograminearum* in the stem base of wheat at 10, 20, and 35 dpi was found to be 1.18555 × 10^−6^ ng, 2.11905 × 10^−6^ ng, and 1.01956 × 10^−5^ ng, respectively. In the *P. chrysosporium* treatment group, the DNA content of *F. pseudograminearum* in wheat roots at 10, 20, and 35 dpi was determined to be 5.06 × 10^−7^ ng, 7.73358 × 10^−7^ ng, and 9.42861 × 10^−7^ ng, respectively. The DNA content of *F. pseudograminearum* in the stem base of wheat at 10, 20, and 35 dpi was measured to be 5.12512 × 10^−7^ ng, 5.86567 × 10^−7^ ng, and 4.26819 × 10^−6^ ng, respectively. In contrast to the DNA content of wheat root tissues in the pathogen group, the DNA content of *F. pseudograminearum* at 10 dpi, 20 dpi, and 35 dpi of *P. chrysosporium* treatment group exhibited reductions of 73.17%, 76.77%, and 68.22%, respectively. Similarly, when compared to the DNA content of wheat stem base tissues in the pathogen group, the DNA content of *F. pseudograminearum* in wheat stem base tissues at 10 dpi, 20 dpi, and 35 dpi in the *P. chrysosporium* treatment group decreased by 56.77%, 72.32%, and 58.14%. These findings suggested that the utilization of *P. chrysosporium* led to a decrease in the DNA content of *F. pseudograminearum* within wheat tissues, consequently resulting in a reduction in the population of *F. pseudograminearum*.

## 4. Discussion

Fungal biocontrol agents can employ a range of direct and indirect strategies to combat pathogenic fungi, including the production of fungistatic agents, mycoparasitism, competition for nutrients, the stimulation of plant resistance to pathogens, and the enhancement of plant growth. The practice of co-culturing takes advantage of the specialized capacities that fungi have evolved to thrive in environments that also sustain high concentrations of other microorganisms. Biocontrol fungi are able to achieve mycoparasitism in co-culture conditions with pathogenic fungi through the secretion of CWDEs, which allows them to obtain partial or complete nutrition from the fungal host [43,44]. These interactions between microorganisms result in the activation of intricate regulatory mechanisms. Mycoparasitism is a prevalent ecological interaction wherein one fungus parasitizes another living fungus. Throughout this interaction, the mycoparasite detects a potential host, exhibits directional growth toward it, attaches to the host, and subsequently forms infection structures to facilitate host penetration. It has been shown that mycoparasites play an essential role in biocontrol fungi to fight against plant fungal diseases. Several mycoparasites, including *Trichoderma harzianum*, *Clonostachys rosea*, etc. [34,45] have been reported. In this study, *P. chrysosporium* showed a variety of ways to inhibit the pathogen *F. pseudograminearum*. The co-culture in our experiment revealed the direct antagonistic activity of *P. chrysosporium* against *F. pseudograminearum* on plates through growth space competition and mycoparasitism. In the mycoparasitic process, a variety of CWDEs were excreted to facilitate the digestion and elimination of pathogen cells.

Secondary metabolites from *Bacillus subtilis*, Actinomycetes, and fungal species were studied for their antifungal properties [46,47,48]. It has been reported that *B. halotolerans* can produce antifungal VOCs, targeting a variety of phytopathogenic fungi [49]. Moreover, VOCs from ectomycorrhizal fungi and arbuscular mycorrhizal fungi can also directly inhibit phytopathogenic fungi or indirectly enhance plant resistance against pathogens in soil [50]. Other fungi like *Trichoderma* sp. have also been reported to exhibit multiple antagonistic mechanisms at the same time [51]. The antagonistic effects of secondary metabolites, including VOCs and fermentation byproducts, produced by *P. chrysosporium* on *F. pseudograminearum* growth were also confirmed. Interestingly, the addition of *F. pseudograminearum* cell wall preparations led to a significant increase in the activity of CWDEs (proteases, chitinases, endoglucanases, and β-1,3-glucanases) in the induced fermentation broth of *P. chrysosporium*, suggesting better antagonistic effects during the co-cultivation process.

To further understand the possible mycoparasitic mechanisms of *P. chrysosporium* and the potential active secondary metabolites, we performed transcriptome analyses. The analysis of the co-transcriptome indicated that during the overgrowth of *P. chrysosporium* with *F. pseudograminearum*, there was a notable enrichment of up-regulated genes. A cluster analysis of genes associated with the potential antifungal mechanisms of *P. chrysosporium* revealed a significant up-regulation in the transcription levels of genes involved in CWDEs, mycotoxin synthesis genes with antifungal properties, and iron ion binding activity associated with nutrient competition. As a white-rot fungus, *P. chrysosporium* had the ability to degrade polysaccharides in lignified plant cell walls. Most proteins corresponding to plant CWDEs (cellulolytic, hemicellulolytic, pectinolytic, esterase, and auxiliary activity) were identified in cultures of *P. chrysosporium* [52]. As a mycoparasite, *P. chrysosporium* produced robust fungal CWDEs that enabled it to effectively target *F. pseudograminearum*. CWDEs may play a role not only in parasitizing *F. pseudograminearum* but also in shaping the cell wall of *P. chrysosporium* for chlamydospore formation for living environment competition. Iron plays a crucial role in various fungal physiological functions, including DNA replication, transcription, metabolism, and energy production through respiration. Due to the common occurrence of iron limitation in the host environment, pathogens rely on this metal for their virulence, leading to the evolution of complex mechanisms for pathogenic fungi to acquire iron from host reservoirs [53]. After 84 h of confrontation, *P. chrysosporium* exhibited competitive behavior for iron by up-regulating the expression of genes related to iron ion binding, thereby impeding the proliferation of pathogens during the confrontation. Mycotoxins are deleterious secondary metabolites synthesized by fungi in response to external stimuli, which trigger signal transduction pathways that significantly impact fungal physiology. The activation of the entire fungal genome’s transcriptional regulatory machinery, particularly transcription factors governing the expression of genes encoding mycotoxins, was observed. The synthesis of these mycotoxins may play a role in pathogenicity or competitive interactions with other organisms for survival. It has been reported that mycotoxins such as gliotoxin and flocculosin can inhibit the growth of pathogens during fungal parasitism processes [54,55]. Our results indicated that in the late stage of fungal confrontation, *P. chrysosporium* up-regulated the transcription levels of genes related to mycotoxin biosynthesis, enhancing the synthesis of mycotoxins, reflecting its potential antifungal activity. However, the responses of *P. chrysosporium* to interactions with *F. pseudograminearum* were examined, highlighting the reciprocal nature of this process. Further investigation into the responses of *F. pseudograminearum* when challenged by *P. chrysosporium* is necessary to elucidate the mechanisms underlying this two-way interaction, offering a valuable opportunity to enhance understanding of biological control mechanisms.

The presence of associations between plants and microorganisms in nature can have varying effects on host plants, ranging from beneficial to detrimental. Encouraging favorable plant–microbe interactions to enhance crop yield and quality represents a promising approach toward achieving eco-friendly and sustainable crop production [56]. The growth-promoting effects of various microorganisms on plants have been explored, including bacteria, fungi, and mycorrhizal fungi [57]. In this study, the impact of *P. chrysosporium* on wheat growth was initially assessed prior to confirming its antagonistic effect on inhibiting pathogens. The results suggested that *P. chrysosporium* treatment yielded significant improvements in various plant growth parameters such as plant height, aboveground fresh weight, root length, and root viability.

In recent years, there has been a growing number of studies discussing the utilization of biocontrol fungi as a means of managing wheat stem base rot, which is attributed to *F. pseudograminearum*. One study demonstrated the antagonistic effects of *Chaetomium globosum* 12XP1-2-3 mycelium and fermentation broth on *F. pseudograminearum*, as evidenced by a delay in disease onset in the stem base and a reduction in disease index. Furthermore, these treatments were found to positively impact wheat yield, enhance the diversity and abundance of beneficial bacteria in the wheat rhizosphere, and suppress the growth of pathogenic fungi in the root vicinity [58]. A study conducted by Zhao et al. [25] identified 157 endophytic fungi isolated from *Cornus officinalis* Sieb, with 17 strains exhibiting antagonistic effects against *F. pseudograminearum*. Among these strains, six demonstrated nutrient competition and five showed antibiotic inhibition. Our research indicated that the utilization of *P. chrysosporium* exhibited significant effectiveness in controlling wheat stem base rot, as demonstrated by a decrease in both disease severity and incidence, surpassing the efficacy of carbendazim as a control measure. The findings from plate culture assays and absolute fluorescence quantitative PCR analyses demonstrated that the presence of *P. chrysosporium* resulted in a reduction in pathogen colonization within the roots and stem bases of wheat. Specifically, *F. pseudograminearum* was restricted to the superficial cells of wheat roots, thereby hindering its progression from the roots to the aerial parts of the plants. This restriction resulted in a significant reduction in disease severity and incidence, ultimately achieving effective disease control.

Plants are capable of perceiving and identifying potential invading microorganisms, whether pathogenic or non-pathogenic, through physical contact in their natural environment. This interaction initiates defense mechanisms that result in the development of induced systemic resistance (ISR) and systemic acquired resistance (SAR). Beneficial microorganisms can establish colonies within host plants, prompting physiological alterations that stimulate the plants’ immune responses and bolster their defenses against pathogenic fungi [59]. It was reported that *T. longibrachiatum* HL167 showed significant salt tolerance and antagonistic activity toward *F. oxysporum* [60]. Additionally, the presence of these microorganisms led to the increased activity of antioxidant enzymes associated with disease resistance, as well as significant up-regulation of defense-related genes (*PR* 2, *CHI* 1, and *PR* 1–2) in seedlings [61]. The application of *P. chrysosporium* led to an increase in the activity of antioxidant enzymes (POD, CAT, and SOD) in plants, an up-regulation of the transcription levels of plant antioxidant enzyme genes, efficient scavenging of oxygen free radicals generated during plant immune responses, and a reduction in the toxicity of *F. pseudograminearum* to plants. Moreover, the utilization of *P. chrysosporium* resulted in elevated expression levels of plant defense-related genes, such as pathogenesis-related (*PR*) genes, β-1,3-glucanase (*Glb* 3), and chitinase (*CHI* 1), which subsequently decreased the accumulation of specific toxin forms in seedlings and induced pathogen tolerance.

In summary, this study systematically investigated the control mechanism of *P. chrysosporium* on wheat crown rot induced by *F. pseudograminearum*. The inhibitory effect of *P. chrysosporium* on the growth of *F. pseudograminearum* and its role in enhancing plant defense against pathogenic fungi were elucidated through physiological and transcriptional analyses. The introduction of *P. chrysosporium* demonstrated positive effects on the plants’ defense mechanisms, including the up-regulation of genes related to disease and stress, the promotion of wheat seedling growth, and a reduction in wheat crown rot incidence. This study highlights the potential of beneficial microorganisms to mitigate the effects of pathogenic fungi and enhance plant immune responses.

Many localities have traditionally employed physical and chemical methods to treat crop diseases, leading to environmental pollution and posing an increasingly serious threat to humanity. In contrast, biological methods offer clear advantages, including ecological compatibility, the absence of secondary pollutant formation, and long-term operational integrity [62]. However, although numerous antagonistic agents exhibit efficacy in laboratory experiments, they are often vulnerable to environmental fluctuations and may not provide sustained protection in practical applications [63]. *P. chrysosporium* has been documented to grow effectively on crop residues, potentially providing sustained efficacy in controlling disease development [64]. Moreover, *P. chrysosporium* could serve as an enduring platform for the microbial modulation of plant health. This study provides novel insights into the biocontrol effects and mechanisms of a white-rot fungus against Fusarium diseases and lays a solid foundation for the development of sustainable approaches to managing wheat crown rot. However, there are notable distinctions between potted and field environments, with the latter demonstrating greater complexity. It is imperative to assess the feasibility of utilizing biocontrol agents in field habitats, assess the efficacy of control strategies, and investigate the potential colonization and persistence of biocontrol fungi in field soils.

## Figures and Tables

**Figure 1 jof-10-00641-f001:**
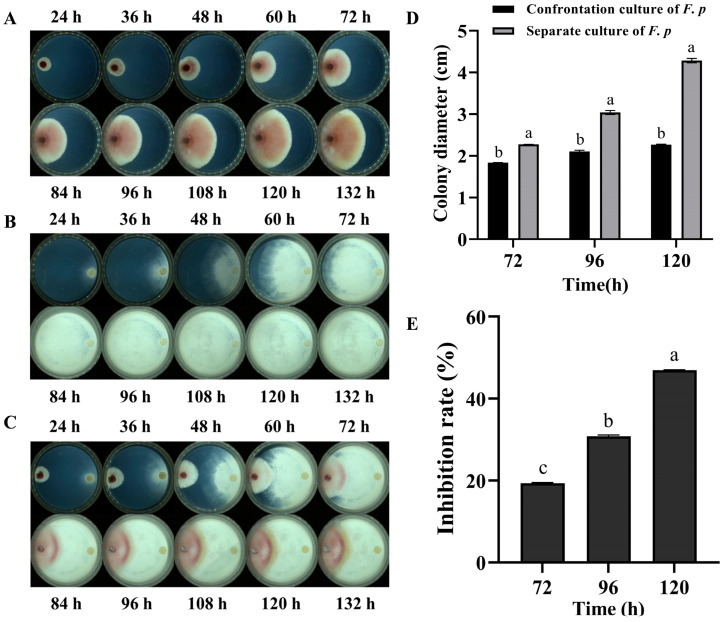
Plate confrontation experiment. (**A**) *F. pseudograminearum* (*F. p*) was cultured alone. (**B**) *P. chrysosporium* (*P. c*) was cultured alone. (**C**) The left side of the confrontation group was *F. p*, and the right side was *P. c*. The mycelial growth of each group was observed at 24 h, 36 h, 48 h, 60 h, 72 h, 84 h, 96 h, 108 h, 120 h, and 132 h. (**D**) Changes in the colony growth diameter of *F. p* in confrontation culture and single culture at 72 h–120 h. (**E**) Inhibition of *F. p* growth by *P. c.* The data are expressed as mean ± SD of three replicates. Different lowercase letters (a, b, c) above bars indicate significant differences at *p* < 0.05 (Duncan’s multiple range test).

**Figure 2 jof-10-00641-f002:**
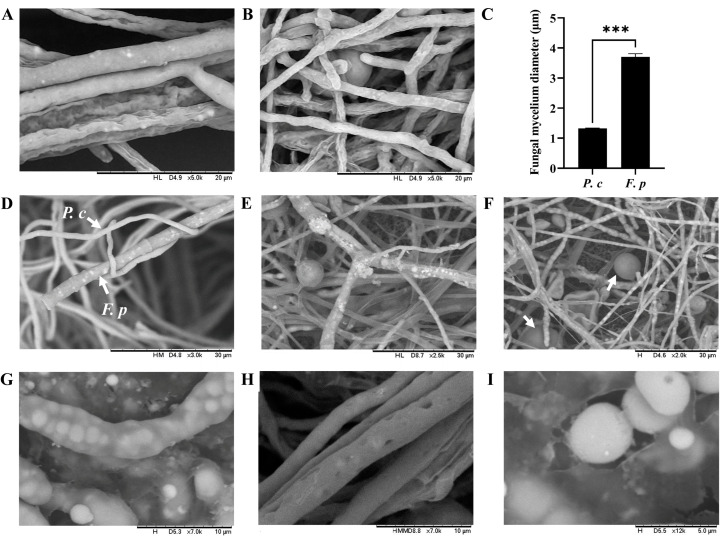
Direct mycelium interaction of *P. chrysosporium* (*P. c*) with *F. pseudograminearum* (*F. p*) under a laser scanning confocal microscope (LSCM) at magnifications ranging from ×2000 to ×12000: (**A**) Mycelium of *F. p*. (**B**) Mycelium of *P. c*. (**C**) Comparison of mycelial diameter between *P. c* and *F. p*. (**D**) *P. c* entwined with and adhered to *F. p.* (**E**) The emergence of an adsorption structure from *P. c* on *F. p*. (**F**) Chlamydospore formation by *P. chrysosporium* to compete for growth space; the arrow indicates chlamydospores. (**G**–**I**) Mycelial rupture in *F. p.* The data are expressed as mean ± SD of three replicates. *** Indicates significant differences between treatments at *p* < 0.001 (*t*-test).

**Figure 3 jof-10-00641-f003:**
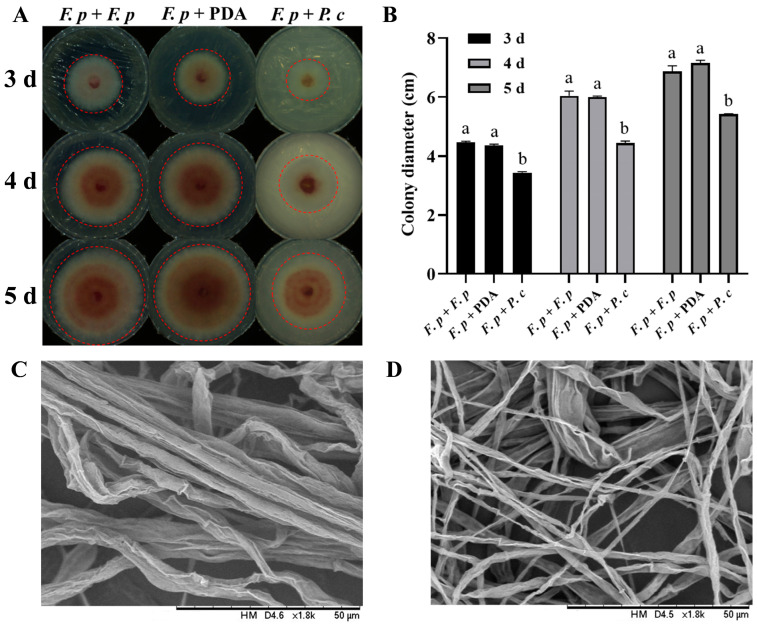
Results of the antifungal experiment with volatile organic compounds (VOCs) from *P. chrysosporium* (*P. c*) against *F. pseudograminearum* (*F. p*). (**A**) Colony growth status of *F. p*–*F. p* confrontation group, *F. p*–*F. p* confrontation group, and *F. p*–*P. c* confrontation group on the 3rd, 4th, and 5th days. Red dotted line area: colony of *F. p*. (**B**) Changes in the colony growth diameter of each treatment group on the 3rd, 4th, and 5th days (*F. p*–*F. p* plate confrontation group, *F. p–*PDA confrontation group, *F. p*–*P. c* confrontation group). (**C**) Mycelial growth state of *F. p* in the *F. p*–*F. p* confrontation group and the *F. p*–PDA plate confrontation group under a scanning electron microscope (SEM). (**D**) Mycelial growth state of *F. p* in the *F. p*–*P. c* confrontation group under SEM. The data are expressed as mean ± SD of three replicates. Different lowercase letters (a, b) above bars indicate significant differences at *p* < 0.05 (Duncan’s multiple range test).

**Figure 4 jof-10-00641-f004:**
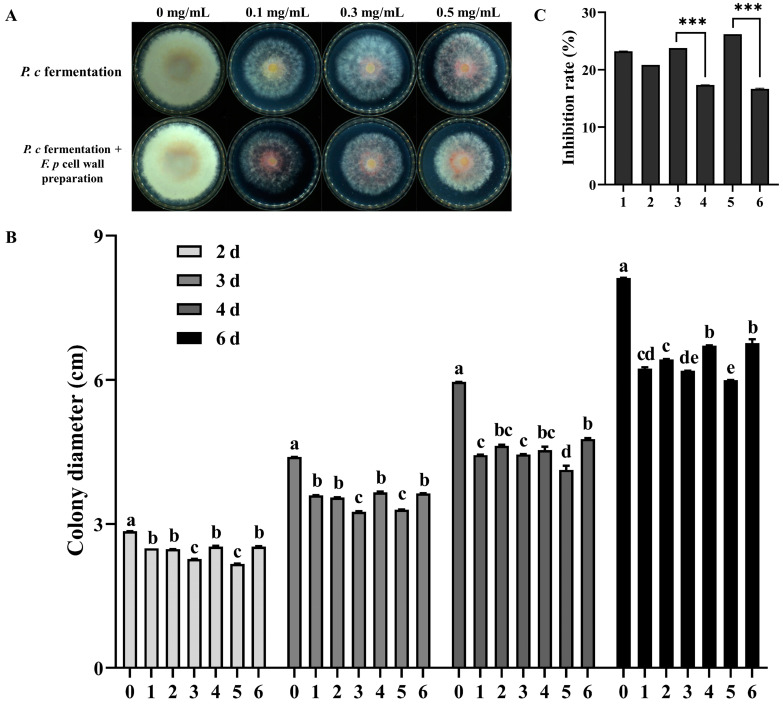
Inhibitory effect of fermentation products and induced fermentation products (with *F. p* cell wall preparation) from *P. chrysosporium* (*P. c*) on *F. pseudograminearum* (*F. p*) and the cell-wall-degrading enzyme activity in induced *P. c* fermentation filtrate. (**A**) Effect of different concentrations of *P. c* fermentation broth and induced fermentation broth on the growth of *F. p*. (**B**) Changes in the colony diameter of each treatment group, of which 0 is the control group (PDA) and 1, 2, and 3 are the changes in colony diameter of the fermentation broth group inoculated only with *P. c* fermentation products, respectively. Labels 4, 5, 6 are the changes in colony diameter in the group inoculated with induced fermentation products. (**C**) Statistics of the colony growth inhibition rate in each group. Numbers 1 through 6 are consistent with the aforementioned descriptions. The data are expressed as mean ± SD of three replicates. *** Indicates significant differences between treatments at *p* < 0.001 (*t*-test). Different lowercase letters (a, b, c, d, e) above bars indicate significant differences at *p* < 0.05 (Duncan’s multiple range test).

**Figure 5 jof-10-00641-f005:**
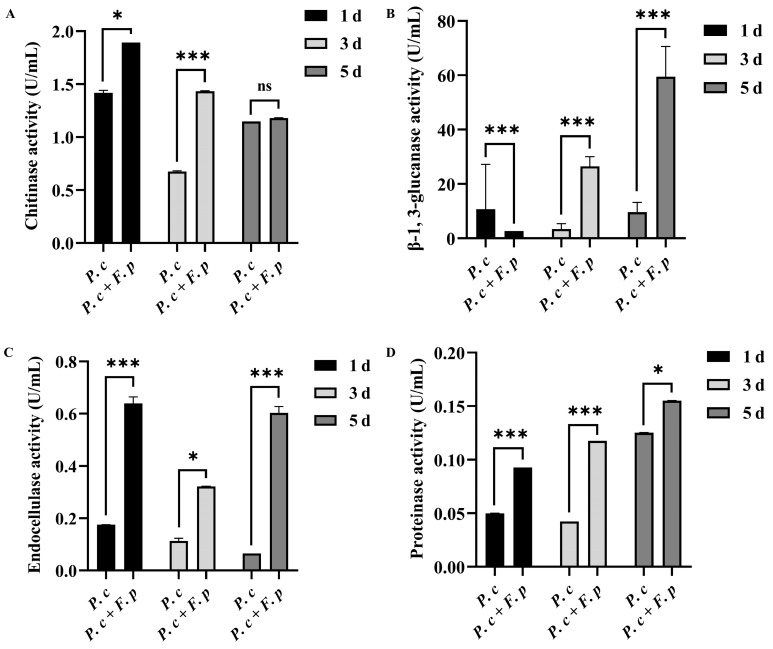
Chitinase, β-1,3-glucanase, endoglucanase, and protease activities in fermentation filtrate, *P. chrysosporium* (*P. c*) fermentation products, and *P. chrysosporium* (*P. c*) + *F. pseudograminearum* (*F. p*) induced fermentation products. (**A**) Chitinase activity. (**B**) β-1,3-glucanase activity. (**C**) Endoglucanase activity. (**D**) Protease activity. The data are expressed as mean ± SD of three replicates. * and *** indicate significant differences between treatments at *p* < 0.05 and *p* < 0.001, respectively (*t*-test). ns: Means followed by the same letter do not differ statistically from each other in the *t*-test.

**Figure 6 jof-10-00641-f006:**
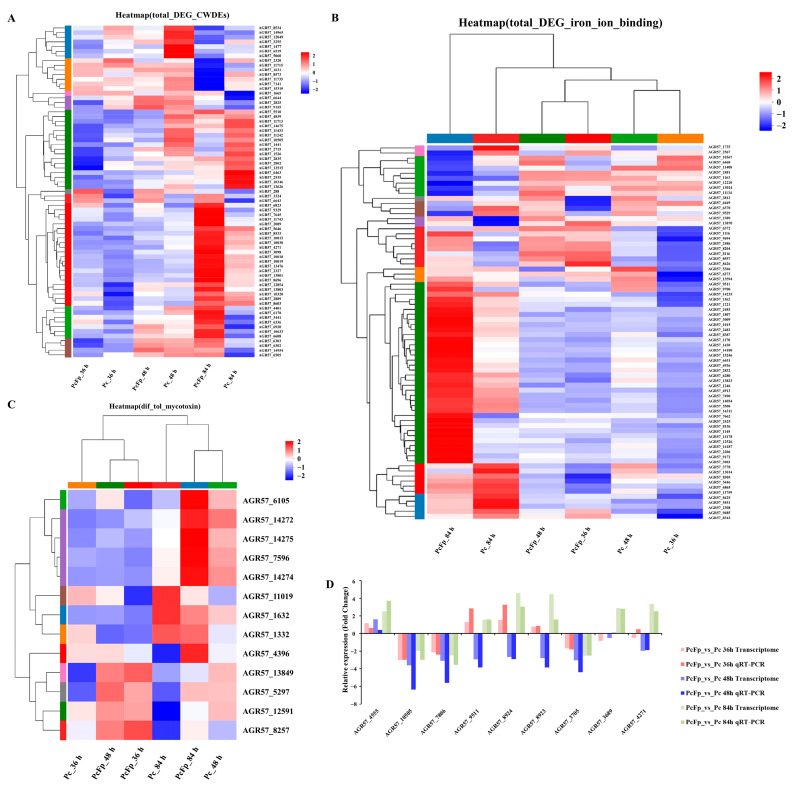
The statistical analysis of differentially expressed genes (DEGs) in *P. chrysosporium* in confrontation with *F. pseudograminearum* was conducted, followed by confirmation of the gene expression pattern of the DEGs using quantitative real-time polymerase chain reaction (qRT-PCR). (**A**) Expression of CWDEs in *P. chrysosporium* detected during mycoparasitism. (**B**) Expression of iron ion binding in *P. chrysosporium* detected during mycoparasitism. (**C**) Expression of mycotoxins in *P. chrysosporium* detected during mycoparasitism. (**D**) Confirmation of gene expression patterns in *P. chrysosporium.* The data are expressed as mean ± SD of three replicates.

**Figure 7 jof-10-00641-f007:**
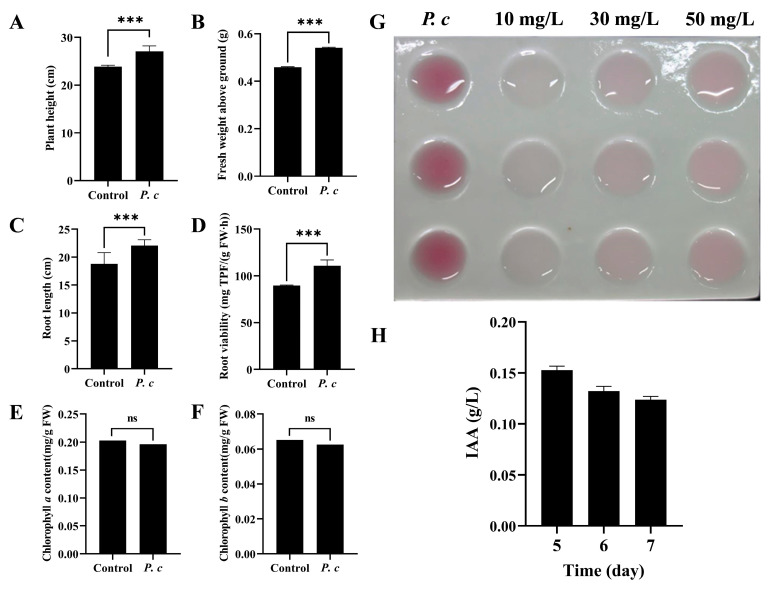
Effects of *P. chrysosporium* (*P. c*) inoculation on wheat growth and IAA-producing ability of *P. chrysosporium*. (**A**) Effects of *P. c* inoculation on plant height of wheat. (**B**) Effects of *P. c* inoculation on the fresh weight of aboveground parts of wheat. (**C**) Effects of *P. c* inoculation on the root length of wheat. (**D**) Effect of *P. c* inoculation on the root viability of wheat. (**E**) Effect of *P. c* inoculation on the chlorophyll *a* content of wheat leaves. (**F**) Effect of *P. c* inoculation on the chlorophyll *b* content of wheat leaves. (**G**) Colorimetric assay using the Salkowski reagent method to detect the presence of IAA. (**H**) Changes in IAA content in culture medium at different culture times. The data are expressed as mean ± SD of three replicates. *** Indicates significant differences between treatments at *p* < 0.001 (*t*-test). ns: Means followed by the same letter do not differ statistically from each other for the *t*-test.

**Figure 8 jof-10-00641-f008:**
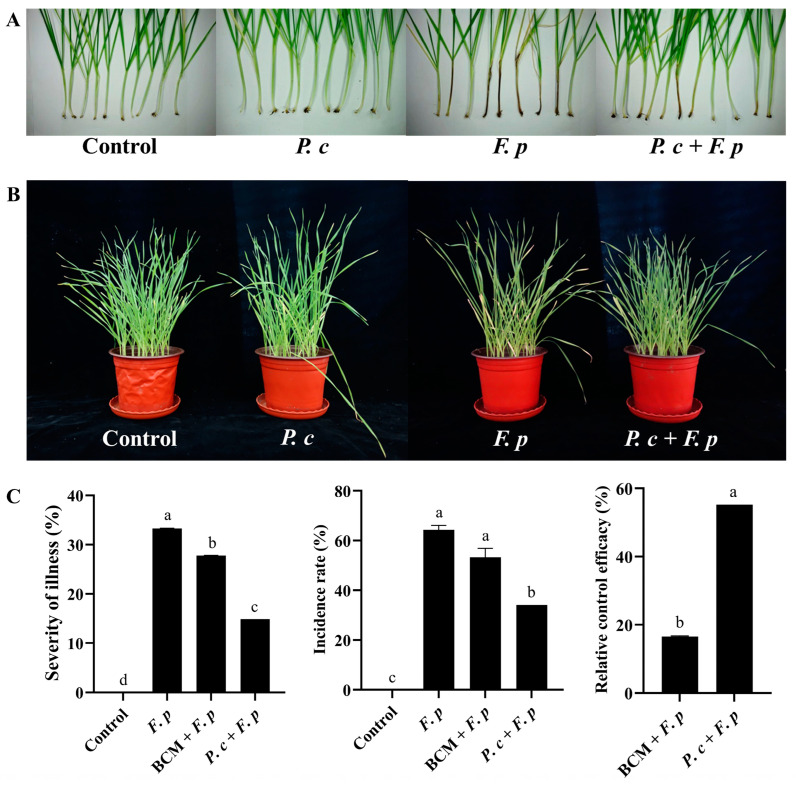
Pot experiment in controlling wheat crown rot using *P. chrysosporium* (*P. c*) at 25 dpi. (**A**,**B**) Impact of different treatments (control group, *P. c* group, *F. p* group, and *P. c* + *F. p* treatment group) on wheat stem base browning and plant growth status. (**C**) Efficacy of *P. c* treatment and carbendazim (BCM) treatment in controlling wheat disease incidence and severity and relative control efficiency. The data are expressed as mean ± SD of three replicates. Different lowercase letters (a, b, c, d) above bars indicate significant differences at *p* < 0.05 (Duncan’s multiple range test).

**Figure 9 jof-10-00641-f009:**
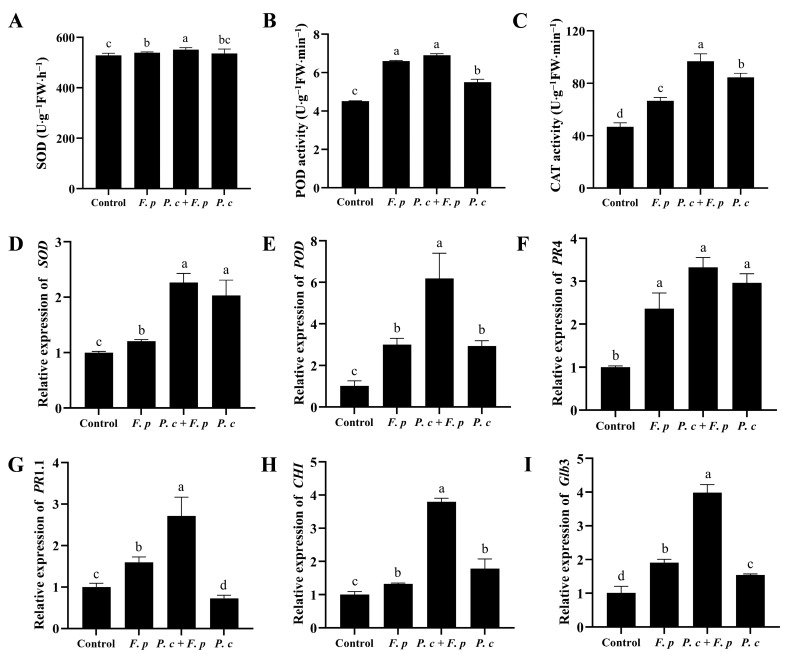
Activity of antioxidant enzymes, along with the transcriptional regulation of antioxidant enzymes and defense-related genes, in wheat subjected to various treatments. (**A**) Superoxide dismutase (SOD). (**B**) Peroxidase (POD), (**C**) Catalase (CAT). (**D**) *SOD* gene. (**E**) *POD* gene. (**F**) Pathogen-associated protein 4 (*PR* 4). (**G**) Pathogen-related protein 1.1 (*PR* 1.1). (**H**) Chitinase 1 (*CHI*). (**I**) β-1,3-glucanase precursor gene (*Glb* 3). The data are expressed as mean ± SD of three replicates. Different lowercase letters (a, b, c, d) above bars indicate significant differences at *p* < 0.05 (Duncan’s multiple range test).

**Figure 10 jof-10-00641-f010:**
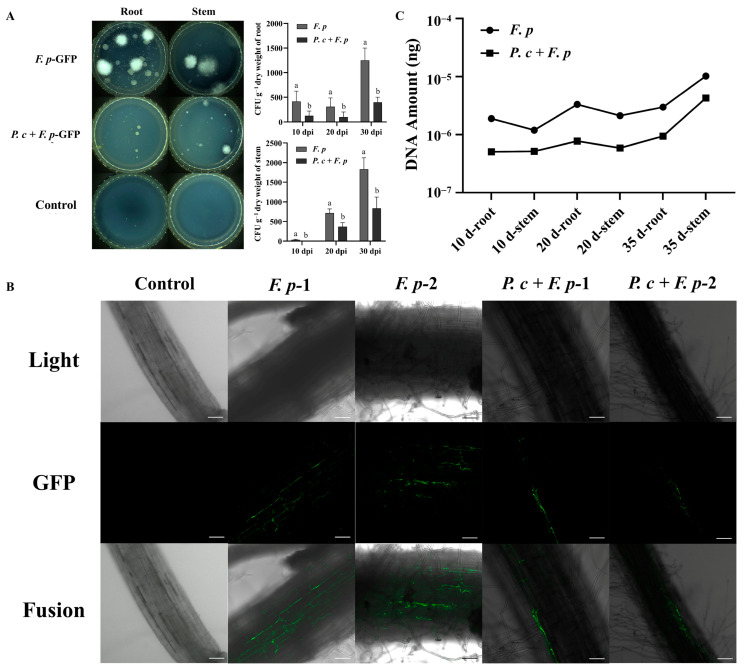
*P. chrysosporium* (*P. c*) reduced the colonization of *F. pseudograminearum* (*F. p*) in wheat. (**A**) The findings of the plate separation experiment conducted on the roots and stems of the diseased portion at 30 dpi are presented, along with the statistical analysis of pathogen colonization in the root and stem base of wheat at 10, 20, and 30 dpi. (**B**) The colonization of *F. p* in wheat roots of different treatment groups was observed by LCSM at a magnification of ×200. (**C**) DNA content of *F. p* in wheat roots and stem base tissues of different treatment groups at 10, 20, and 30 dpi. The data are expressed as mean ± SD of three replicates. Different lowercase letters (a, b) above bars indicate significant differences at *p* < 0.05 (Duncan’s multiple range test).

## Data Availability

The original contributions presented in the study are included in the article/Appendix A, further inquiries can be directed to the corresponding author.

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
