# Peer review of "Exploring the Biocontrol Potential of Phanerochaete chrysosporium against Wheat Crown Rot"

_jof, 2024, doi:10.3390/jof10090641_

Round 1

Reviewer 1 Report

REVIEW REPORT

General concept comments

Article:

This paper refers to the possibility of employing Phanerochaete chrysosporium against wheat crown rot. The submitted manuscript is a fairly well-organized theoretical paper. However, there are several important issues regarding methods and presentation of results, which should be addressed before the work can be published.

It should be noted that all comments, questions and suggestions below are presented with the sole intention of improving the quality of the manuscript for possible publication, and not in a personal capacity.

The following recommendations should be followed for the article to be published.

Major comments:

·    The last paragraph of the introduction is very critical. The authors have summarized the key area of research, and the aim of the paper but not the novelty of the communication.

·   Have duplicates or triplicates of the developed plate assays been performed? Please add the deviation of the different assays such as inhibition rate and colony diameter. 

· Conclusions shall rewrite by focusing on new contributions when compared with the previous works. Key values, environmental consequences, and future work activities should be mentioned in the conclusions as well.

REVIEW REPORT

Detail comments:

·     Please use the Microsoft Equation Editor or the MathType add-in to define the equations described throughout the manuscript (i.e; lines 115, 239…). They are complex to understand because they are mixed with the text. Remember that equations must be editable by the editor and not appear in image format.

·     Figure 4 (line 384): There are too many subfigures in this section. It is necessary to divide it into two figures to improve its visualization. It is impossible to visualize the data in Figure 4.A and 4.B. The lettering is too small.

·         Figure 5 (line 443) : Like Figure 4, the text in Figures 5A, 5B and 5D is illegible. Please redo considering that the figure should be completely legible.

Reviewer 2 Report

Please explain with more details,  what was the doses criteria to use  1x10cells in  the experiment

Please in the discussion section can you explain with more details the pontential use of this microorganins

please the the author can make more great the figure 4

Reviewer 3 Report

These studies are relevant in the field of existing biocontrol in the processes and technologies of wheat cultivation, as the main food crop for humanity. Any biological object and biological environment under study is a difficult object to study. The variability and variability of biological objects requires the use of new scientific approaches for biocontrol, which are proposed by the authors in this article.  In studies, there is a difference in values ​​between laboratory and field studies. When conducting research in the laboratory, the authors limited a number of variable values, which simplified the obtaining of significant indicators but also led to a correlation between the closed environment of wheat cultivation and field technologies for their cultivation. Wishes to the authors to conduct this research. In the field environment directly with standard wheat cultivation technologies.

- Line 47 Expand the list of pathogens. The causative agents of root rot of wheat are also fungi of the genus Fusarium (F. culmorum, F. avenaceum, F. oxysporum).

(Ahmed Saad etal. (2023) https://doi.org/10.1016/j.cj.2022.08.013; Yiyi Syonget (2023) et al.  https://doi.org/10.1111/ppa.13777)

- Lines 64-65 Provide references to literature

- Line 101 Indicate from which isolated plant materials it was obtained 

P. chrysosporium (ATCC20296), and also plays a role in micromycete isolation techniques.

- Line 247. …… with an inhibition rate of 47% (Fig. 1E). The size in the text must correspond to the size in the figure. 1E (0.4%?)

Reviewer 4 Report

The research topic is related to the analysis of the mechanisms of biocontrol action of Phanerochaete chrysosporium against phytopathogen of wheat crown rot Fusarium pseudograminearum, causing serious damage to wheat productivity. The authors investigated the mycelial growth dynamics of both fungal species under co-cultivation, revealed the inhibitory effect of volatile organic compounds of mycoparasitic fungus on the growth of pathogenic fungus. They determined that the decrease in pathogen viability was accompanied by up-regulation of differentially expressed genes (DEGs) encoding fungal cell-wall-degrading enzymes, iron ion binding and mycotoxins. In addition, the reduction in disease severity and incidence of wheat as well as the plants growth promotion due to the antagonistic interactions of fungi and induction of plant immunity were shown in a pot experiment with plants, inoculated with F. pseudograminearum.

The study represents a well-conducted work with the application of various methods revealing the molecular-genetic, biochemical, and environmental mechanisms of interaction between the studied fungal strains. The literature review and discussion is based on recent scientific articles, 65% of which were published in the last 5 years. The methods are described in sufficient detail to be able to reproduce them. The presented photo documents and well-organized data of the research results convincingly support the hypothesis put forward in the research objectives about the possibility of using of wheat plants inoculation with P. chrysosporium spore solution in agrobiotechnology. The results of the study contribute to the understanding of fundamental aspects of the interaction of organisms in the fungal community, and also contribute to the expansion of the arsenal of biological plant protection products, in particular for organic farming. 

During the manuscript reading, several questions and comments arose:

1) In the sentence (lines 12-14) the expression “developing specialized structures for penetration of the pathogen” (line 14) may need to be replaced as it introduces a semantic misleading.

2) The sections 2.6 “The impact of P. chrysosporium on the growth of wheat” should be written in the past tense in the passive voice.

3) It is advisable to indicate how many plants were taken for growth and disease analysis.
